# Case Reports for Topical Treatment of Corneal Ulcers with a New Matrix Therapy Agent or RGTA^®^ in Dogs

**DOI:** 10.3390/vetsci6040103

**Published:** 2019-12-13

**Authors:** Jessica A. Martinez, Franck Chiappini, Denis Barritault

**Affiliations:** 1Eye Center for Animals Inc., 524 Moss Street, New Orleans, LA 70119, USA; jessica.martinez.v12@gmail.com; 2Animal Eye Guys, Fort Lauderdale, Florida, FL 33308, USA; 3Organ, Tissue, Regeneration, Repair and Replacement Company (OTR3), F-75001 Paris, France; franck.chiappini@otr3.com; 4Laboratoire Croissance, Régénération, Réparation et Régénération Tissulaires (CRRET), EAC CNRS 7149, University Paris Est Creteil, F-94010 Créteil, France

**Keywords:** regenerative medicine, Clerapliq^®^, regenerating agent, ophthalmology, otr4120

## Abstract

Superficial corneal ulcers that fail to heal within a normal time period and are refractory to conventional therapy in dogs are common in veterinary practice. Different etiologies can lead to this result, including spontaneous chronic corneal epithelial defects (SCCEDs) and ulcerative keratitis associated with bullous keratopathy. Thus, there is an urgent need to find new therapeutic approaches such as matrix therapy replacement. To determine the efficacy of a new ophthalmic treatment (Clerapliq^®^) for SCCEDs and ulcerative keratitis associated with bullous keratopathy, a total of 11 dogs referred to the clinic because of nonhealing erosive ulcers after a classic primary treatment were enrolled to get this new treatment. Dogs underwent ophthalmic exams and 7 dogs (10 eyes) were diagnosed with superficial ulceration and 4 dogs (5 eyes) with bullous keratopathy due to endothelial dystrophy/degeneration. They received eye drops of Clerapliq^®^ every 3 days until recovery. The results showed that the corneas with recurrences of the ulcers were resolved predominantly by using Clerapliq^®^ every 3 days in 83.3% of the cases during a period of treatment ranging between 6 to 35 days. Therefore, this new approach using matrix therapy regenerating technology in treating superficial ulcers and bullous keratopathy in dogs can be successfully considered as an adjunctive therapy.

## 1. Introduction

Over the past few years, a new type of matrix therapy agent named ReGeneraTing Agent (RGTA^®^) has provided encouraging results, accelerating the healing of chronic skin ulcers of diabetic or vascular origin [1,2,3,4]. RGTA^®^ is a set of molecules, chemically engineered polymers, that are specifically designed to replace degraded heparan sulfate molecules in the injured matrix compartment. Therefore, they are considered as heparan sulfate mimetics based on their chemical structures and functions [3,4,5,6]. RGTA^®^ protects naturally existing structural and signaling proteins, and in doing so, creates a cellular microenvironment favorable to healing, thereby enhancing the speed and quality of tissue repair [4,7].

Indeed, the importance of extracellular matrix (ECM) integrity in maintaining normal tissue function is highlighted by numerous pathologies and situations of acute and chronic injury associated with dysregulation or destruction of ECM components. Heparan sulfates (HS) are a key component of the ECM, where they fulfill important functions associated with tissue homeostasis. HS belong to the glycosaminoglycan (GAG) family. Their degradation following tissue injury disrupts this delicate equilibrium and may impair the healing process. RGTA^®^ are specifically designed to replace degraded HS in injured tissues. The unique properties of RGTA^®^, such as resistance to degradation, binding, and protection of ECM structural and signaling proteins, permit the reconstruction of the ECM, restoring both structural and biochemical functions to this essential substrate, and facilitating the processes of tissue repair and regeneration [4].

Numerous situations implicating acute and chronic injuries are associated with dysregulation or destruction of ECM components. Heparan sulfates are the key components of the ECM, which tightly control tissue homeostasis. OTR4120 is a polysaccharide specifically designed (substituted with carboxymethyls and sulfates) to replace degraded heparan sulfates in injured tissues. The unique properties of OTR4120, as described above, permit the reconstruction of the ECM, restoring the homeostasis of the tissue, facilitating the processes of tissue repair and regeneration [1,4,8,9,10].

Superficial corneal ulcers that fail to heal within a normal time period and are refractory to conventional therapy are common in veterinary practice. Different etiologies can result in refractory corneal ulcers, such as morphologic and neurologic abnormalities of the eyelids, thick distichiasis or ectopic cilia, tear film abnormalities, deficiencies of corneal innervation, foreign bodies, microbial infection, spontaneous chronic corneal epithelial defects (SCCED), and ulcerative keratitis associated with bullous keratopathy [11].

SCCED are superficial epithelial defects that have not become infected, do not involve the corneal stroma, are bordered or partially covered with non-adherent epithelium, and fail to heal in a normal time period. The pathological mechanisms are not fully understood. Poorly adherent epithelium and epithelial dysmaturation at the periphery of lesions with varying degrees of leukocyte infiltration are currently associated with ECM disruption [12]. The basement membrane is typically absent or present in discontinuous segments within the lesion. A hyaline acellular zone in the anterior corneal stroma is commonly present in the area of the erosion [13]. Stromal fibroplasia, vascularization, and leukocyte infiltration have been observed in some specimens. Disorganized zones of sub-epithelial and epithelial hyperinnervation surround the epithelial defect. Matrix metalloproteinase (MMP) activity is elevated in affected corneas and epithelial–mesenchymal transition, the process in which anchored epithelial cells transform into migrating fibroblast-like cells to re-epithelialize corneal epithelial defects, is abnormal [14,15]. In summary, most dogs with SCCED do not have a normal basement membrane structure in the region of the epithelial defect and have other abnormalities in the subjacent extracellular matrix that may reflect a part of the underlying pathophysiology of chronic and recurrent erosions [12].

Bullous keratopathy is a sequela to severe or chronic corneal edema. Endothelial cell dysfunction (either degeneration or dystrophy) and the resultant corneal edema can lead to the formation of intra-epithelial or sub-epithelial bullae. These bullae are at risk of corneal ulceration due to structural weakening of the cornea or spontaneous rupture. These corneal ulcers tend to follow a prolonged healing course and reoccurrence is common [16,17,18].

Because the ECM is altered during the progression of both the bullous keratopathy and the SCCED, multiple treatment modalities have been considered. The management of SCCED treatments include medical therapies such as polysulfated glycosaminoglycans (PSGAGs), protease MMP inhibitors, topical epidermal growth factors, fibronectin and surgical therapy with epithelial debridement, especially diamond burr debridement (DBD), which is now becoming the standard but associated to medical therapies to enhance the healing process [15,19,20]. Indeed, DBD induces mechanical disruption that exposes the proteins of the ECM to reactivate the healing process, allowing exposure of the normal peripheral basal membrane (BM) to the growth of newly formed epithelium, enhancing its adhesion properties [13]. Indeed, removal of the epithelial BM during DBD enhances many wound healing processes in the cornea, including keratocyte apoptosis and nerve death. In addition, the BM controls cellular functions by binding and modulating the local concentrations of growth factors and cytokines, and is able to regulate cell polarity, cell adhesion, spreading, and migration via its effects on the cytoskeleton. [18]. Bullous keratopathy management approaches include mainly medical therapies, and different molecules have been proposed to restore the ECM, such as collagen cross-linking [21,22,23] or chondroitin sulfate [24]. The conclusions of these studies have not been conclusive or convincing. Thus, there is an urgent need to find new therapeutic approaches to replace the injured ECM in these pathologies.

In the domain of ophthalmology, an RGTA^®^ family compound named OTR4120, a heparan sulfate mimetic, has been reported to show encouraging results for the treatment of corneal ulcers and dystrophies of various etiologies [10,25,26]. Furthermore, OTR4120 was described in a case report concerning one patient with a neurotrophic ulcer [26]. During the last few years, OTR4120 eye drops have been successfully used for the treatment of resistant corneal neurotrophic ulcers [8], as well as for the treatment of keratoconus in humans [9]. OTR4120 has been available on the European market for veterinary use as Clerapliq^®^ for more than 5 years [8,9,25,26,27]. The mode of action of OTR4120 is to replace the destroyed HS and restore the ECM scaffold by organizing the collagens in the matrix and by protecting the growth factors and cytokines. This leads to the recreation of the physiological environment required for tissue repair and regeneration [3,4].

The aim of these case reports was to evaluate the beneficial impact of OTR4120 (Clerapliq^®^) to treat primary or secondary corneal epithelial erosion in dogs in the current ophthalmology veterinarian practice. The results show that improvement appeared in 6–35 days, with an average of 18.8 days of treatment once every 3 days (i.e., q3d) among the 11 dogs treated with Clerapliq^®^ simultaneously with the usual treatment. Failure was observed only in 2 dogs due to other complications. In conclusion, these case reports show that Clerapliq^®^ was successful for the treatment of primary or secondary epithelial erosion associated with the etiology treatment. Clerapliq^®^ is indicated for analgesic purposes and for optimizing the time-space healing process of the cornea observed in 83.3% of the cases. Finally, Clerapliq^®^ was easy and safe to use for the practitioner and the owner.

## 2. Materials and Methods

### 2.1. Dogs

A total of 11 dogs were presented at the Eye Center for Animals Inc. (524 Moss Street, New Orleans, LA, USA) clinic for nonhealing corneal ulcers after failure of primary treatment by the referring veterinarian, usually by epithelial debridement, without evidence of defect resolution for several weeks to months. The dogs were then referred to the Eye Center for Animals. A total of 11 dogs (15 eyes) were evaluated in these case reports. All dogs underwent complete clinical evaluation and examination of both eyes. All dogs were privately owned pets. Owners reviewed and signed an informed-consent form before samples were collected, as well as consent for the use of Clerapliq^®^. All research was performed in accordance with the Association for Research in Vision and Ophthalmology (ARVO) Statement for the Use of Animals in Ophthalmic and Vision Research. Written informed consent to treat the dogs with Clerapliq^®^ was obtained from all owners when they attended the Eye Center for Animals Inc. (524 Moss Street, New Orleans, LA, USA), in accordance with the relevant Louisiana state and USDA laws and regulations.

### 2.2. Clinical Evaluation

A total of 11 dogs were presented at the Eye Center for Animals Inc. clinic (New Orleans, LA, USA) for nonhealing corneal ulcers after failure of primary treatment by the referring veterinarian. Briefly, superficial corneal ulcers that fail to heal within a normal time period and are refractory to conventional therapy are common in veterinary practice. Different etiologies can result in refractory corneal ulcers: Morphological and neurological abnormalities of the eyelids, thick distichiasis or ectopic cilia, tear film abnormalities, deficiencies of corneal innervation, foreign bodies, microbial infection, spontaneous chronic corneal epithelial defects (SCCED), and ulcerative keratitis associated with bullae. All dogs underwent complete examination of both eyes. To assess the cause(s) of the treatment failure, complete eye exams were performed at the initial visit and then during follow-up. Clinical follow-up also included the evaluation of local side effects such as local edema, inflammation, pain, pruritus, and watery eyes, and systemic side effects such as body weight, food intake, behavior changes, body core temperature, skin rash, edema, and diarrhea.

### 2.3. Clinical Tests

Visual status, menace, and dazzle responses were assessed at the first visit. Palpebral and pupillary light reflexes were also recorded. Globe, adnexa, and conjunctiva status were examined, as well as the presence of discharge.

Then, each compartment of the eye, such as the cornea, anterior chamber, pupil, lens, vitreous, retina, and optic nerve, were examined at the first visit, as well as during follow-up, as follows.

To assess any retinopathy or optical nerve defects, each dog underwent slit-lamp examination (SL 15; Kowa Optimed Europe Ltd., Sandhurst, UK) after using mydriatic drops by assessing the vertical and horizontal cup-to-disc ratios, the visibility of the lamina cribosa, the color of the cup, the area of the vessels, their tortuosity and collateral vasculature, the contour of the neuroretinal rim and the four quadrants thickness, the retinal nerve localization and aspect with the red filter, the disc size along its axis, and the peripapillary development (e.g., normal or atrophic).

Indirect ophthalmoscopy (Omega 500, HEINE Optotechnik, Herrsching, Germany; 2.2 PanRetinal) and 20D lenses (Volk Optical Inc., Mentor, OH, USA) were used to assess the posterior segment.

Intraocular pressure (IOP) was measured using the TonoVet tonometer (TonoVet; ICare, Vantaa, Finland) at the first visit and during the follow-up of the animal.

### 2.4. Schirmer Tear Test

As a crude estimate of tear production, both eyes of each dog were evaluated using Schirmer test strips (Schirmer Tear Test; Intervet/Schering-Plough Animal Health, Roseland, NJ, USA). One test strip was placed in each conjunctival sac at the junction between the lateral one third and medial two thirds of the lower lid for 60 s, which is the standard time used for tear evaluation in dogs. Immediately after the 60-s period, the length of the moistened area of the test strip was measured and recorded in millimeters. Testing was performed before administration of any topical medication.

### 2.5. Fluorescein Stain

A strip containing fluorescein sodium (Acrivet-Veterinary Division, Hennigsdorf, Germany) was wetted with sterile irrigating solution, and 1 drop of fluorescein added to the affected eye. Then, the eye was washed-out with sterile irrigating solution before examination. The epithelial defect of each dog was then measured from epithelial edge to epithelial edge using a pair of calipers accurate to 0.1 mm. Two measurements of the ulcers were made. The first measurement was made at the greatest dimension of the erosion and the second at a 90° angle to the first measurement.

### 2.6. New Regenerative Matrix Treatment

A total of 11 dogs were presented at the Eye Center for Animals Inc. clinic for nonhealing corneal ulcer after failure of primary treatment by the referring veterinarian. All dogs underwent complete examination of both eyes; among the 11 dogs enrolled, 5 underwent a diamond burr debridement at the clinic, and then all dogs received the matrix therapy agent Clerapliq^®^ eye drop every 3 days (1 drop, q3d) until ulcer healing (usually between 1–3 months). The structure, synthesis pathway, and potential applications of the active substance OTR4120 (CAS RN: 227322-59-0) are presented elsewhere [28], and it belongs to ReGeneraTing Agents (RGTA^®^) family. OTR4120 is already used in two commercially available products for the treatment of corneal lesions and chronic ulcers for humans and pets (Cacicol20^®^ and Clerapliq^®^, respectively) in Europe, with more than 50,000 patients and pets treated in the last 5 years [28]. Briefly, Clerapliq^®^ is a solution that contains OTR4120, an alpha 1-6 poly-(carboxymethyl-sufate)-glucose (Figure 1) with 4 mg/mL of dextran diluted in 9 mg/mL of sodium chloride solution. Clerapliq^®^ is presented in a 0.33 mL sterile single-dose and provided by OTR3 (Paris, France).

### 2.7. Statistical Analysis

The comparisons between the proportion of dogs that successfully responded to Cleraplic^®^ was tested by using a chi-square test (χ^2^).

## 3. Results

### 3.1. New Regenerative Matrix Treatment

A new treatment based on matrix regeneration was used to help and accelerate the healing process. Each affected eye received 1 drop of Clerapliq^®^ containing OTR4120 (Figure 1) every 3 days (i.e., q3d) during a mean time of 21.4 ± 15.9 days (i.e., ranging between 7 to 55 days; Table 1).

### 3.2. Treatment and Follow-up

Dogs treated with Clerapliq^®^ met the following criteria: Presence of a nonhealing corneal epithelial defect in one or both eyes for at least 3 weeks without notable progress towards resolution; absence of any identifiable underlying cause of the persistent defect (e.g., normal tear production; absence of lid conformational defects; normal lid function; no clinical evidence of sepsis; and absence of distichiasis, ectopic cilia, and foreign bodies); and absence of any clinical evidence of systemic disease. Only one dog (dog#8) developed local infection in the eye, leading to bacterial culture and an antibiogram test. The corneal ulcer healed very well after gentamicin treatment.

Among the 11 dogs enrolled, 7 dogs (10 eyes) were diagnosed with superficial ulceration and 4 dogs (5 eyes) were diagnosed with bullous keratopathy due to endothelial dystrophy. All animals received Clerapliq^®^ (Table 1).

Among the 15 eyes treated with Clerapliq^®^, only 2 treated eyes failed to positively respond to the treatment (dog#5 and dog#6) and 3 dogs were lost to follow-up (Table 1). When diamond burr debridement was performed, Clerapliq^®^ treatment started between 7 to 15 days after (median = 11 days).

The success of Clerapliq^®^ treatment was observed in 13 among the 15 eyes treated (i.e., 86.6% success) with a mean treatment time of 15.6 ± 9.7 days (6–35 days). Because 3 dogs were lost to follow-up, the success of the treatment remains 10 among the 12 treated eyes, with follow-up leading to 83.3% success with a mean treatment time of 18.7 ± 10.5 days (6–35 days). The comparison of the dogs with success before (0%; 0/12) and after treatment with Clerapliq^®^ (83.3 %; 10/12) showed a significant improvement (χ^2^
*p*-value = 3.5 × 10^−5^).

Finally, as expected, the dogs did not show any local (i.e., edema, inflammation, pain) or systemic (i.e., no loss of body weight; no decrease in food intake, behavior changes, no increase in body core temperature, skin rash, edema, no diarrhea) side effects after Clerapliq^®^ treatment.

The clinical data are summarized in Table 1 and Appendix A.

All of the details for each clinical case can be found at the end of the manuscript in Appendix B.

## 4. Discussion

OTR4120 was tested for the first time in corneal ulcers and severe dystrophies resistant to standard therapies in humans [25]. The product was administered topically once a week for 1 month and resulted in significant reduction in pain, improvement of keratitis, and healing of the majority of corneal ulcers. Several case reports using RGTA^®^ in cases of neurotrophic keratopathy and corneal ulcers since then have indicated a positive effect of the treatment [26,29,30,31]. The effectiveness of RGTA^®^ in corneal neurotrophic ulcers of various primary etiologies was also examined in a larger study with 11 patients, where RGTA^®^ treatment resulted in complete corneal healing in eight patients, with the remaining patients presenting the most severe cases [8]. More recently, the combination of OTR4120 (Cacicol20^®^) with a bandage contact lens in three patients with persistent epithelial defects promoted complete corneal epithelial healing in 4–21 days [28].

Also, dog#5 and dog#6 did not respond successfully to Clerapliq^®^ (see Appendix B). They were treated for 55 and 40 days, respectively, and they still had persistent epithelial erosion. Failure of the treatment might be because of underlying causes such as aging, topical anti-inflammatory or immunomodulation medications, or canine herpes virus-1 (CHV) infection [32,33,34]. However, based on the clinical data, the main failure of the treatment in these two dogs is most likely due to severe edema preventing healing. Interestingly, previous reports have shown that OTR4120 (Cacicol^®^) has antiviral effects and promotes corneal regeneration in herpes neurotrophic ulcers [26,35], suggesting that Clerapliq^®^ treatment might be efficient in CHV-infected dogs.

Also, because 3 dogs were lost to follow-up, we did not know if they relapsed. We assumed they did not relapse, and because of the success of the treatment, they did not show up. Whatever the case, this loss to follow-up did not change the percentage of success (86.6% vs. 83.3% with 3 dogs missing).

In a recent publication, OTR4120 was found to be no more effective than hyaluronic acid, considered as a placebo, for healing SCCED [36]. We believe that hand removal of the hyaline acellular zone in the anterior corneal stroma may be difficult to fully achieve, and for this reason, diamond burr is preferred. Removal of the hyaline acellular zone is a prerequisite for successful SCCED management and is an absolute prerequisite for Clerapliq^®^ efficacy, in order for OTR4120 to reach the heparan binding sites available in the ECM of the wounded cornea and to restore the healing process. Thus, improper hyaline acellular zone removal may explain the lack of efficacy of OTR4120. Also, it should be noted that local immunosuppression medications should be avoided during treatment with Clerapliq^®^.

Finally, the Clerapliq^®^ treatment used in these case reports did not result in any local or systemic side effects in treated dogs. This is in accordance with the safety data obtained from more than thousands of dogs and cats treated by Clerapliq^®^ in Europe (TVM) and with the publications on Clerapliq^®^ [36]. Also, as mentioned in the introduction, the equivalent of Cleraplic^®^ in humans is called Cacicol^®^, which is a medical device that has been successfully used for more than 50,000 patients with no reported local or systemic side effects [37,38,39].

## 5. Conclusions

Taken together, the corneas with recurrences of the ulcer were resolved predominantly by using the matrix therapy agent eye drop Clerapliq^®^ every 3 days during a period ranging between 6 to 35 days with 83.3% success. Local immunosuppression medications should be avoided during the use of Clerapliq^®^ in order to successfully treat recurrence of corneal ulcer. Therefore, this new approach using matrix therapy regenerating technology in treating superficial ulcers and bullous keratopathy in dogs can be successfully considered as an adjunctive therapy.

## Figures and Tables

**Figure 1 vetsci-06-00103-f001:**
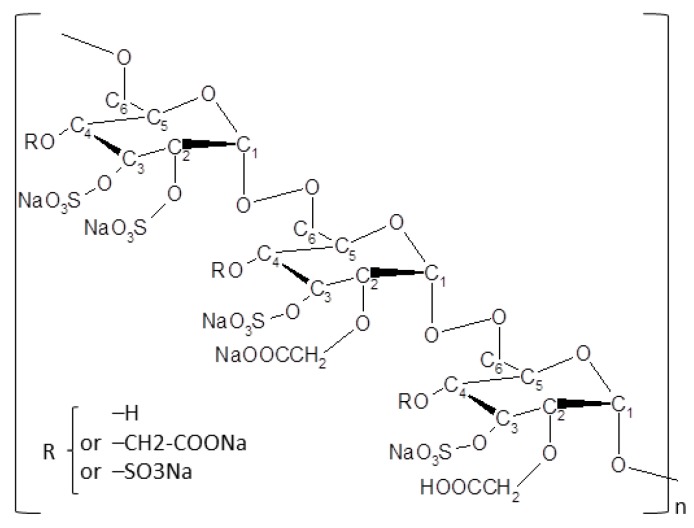
Chemical structure of OTR4120. OTR4120 is a derivatized dextran with carboxymethyl and sulfate substitutions (dextran, hydrogen sulfate, carboxymethyl ether, sodium salt, CAS RN: 227322-59-0), known also as alpha 1-6 poly-(carboxymethyl-sulfate) glucose.

**Table 1 vetsci-06-00103-t001:** Case report populations and clinical data summarized from Appendix B.

Dog #	ID#	Age (years)	Breed	Diagnostic	Eye Treated	Duration of Clerapliq^®^ Treatment (day)	Frequency	Failure (Y/N)	Lost to Follow-up (Y/N)	DBD (Y/N)	Clerapliq^®^ after DBD (day)
**1**	12285	12.8	Shih-Tzu	OS healed anterior stromal ulcer	OS	17	1 drop q3d	0	0	0	0
**2**	13204	10.3	Boston Terrier	Corneal endothelial dystrophy and persistent epithelial erosion	OD	35	1 drop q3d	0	0	1	11
**3**	13256	7.2	Canine	Bullae in temporal paracentral region/persistent epithelial erosion	OD	6	1 drop q3d	0	0	1	9
**4**	13206	9.3	Griffon	Corneal endothelial dystrophy OU/persistent epithelial erosion	OS	27	1 drop q3d	0	0	1	13
**5**	13201	15.4	Chihuahua	OU corneal endothelial dystrophy prognosis: good keratitis and erosions	OU	55	1 drop q3d	1	0	0	0
**6**	13250	13.7	Mix	OU corneal endothelial dystrophy with risk of recurrent erosion; OD corneal epithelial erosion	OD	40	1 drop q3d	1	0	1	15
**7**	13266	11.6	Boston Terrier	Corneal endothelial dystrophy with predisposition to persistent erosion	OD	13	1 drop q3d	0	1	0	0
**8**	13194	13.8	Canine	Post-op phacoemulsification and lens implant for diabetic cataract and persistent corneal erosion	OU	14	1 drop q3d	0	0	0	0
**9**	13241	15.3	Miniature Poodle	OD post-op phacoemulsification and lens implantation; OS hyper-mature cataract and lens induced uveitis; OU severe iris atrophy, suspected nocturnal lagophthalmia and epithelial erosion	OU	13	1 drop q3d	0	0	0	0
**10**	12840	12.1	Pomeranian Mix	OS epithelial erosion, immature cataract	OS	8	1 drop q3d	0	1	0	0
**11**	11031	14.3	Dachshund	OU post-op phacoemulsification and lens implant, corneal endothelial dystrophy, PRA; OD persistent epithelial erosion	OD	7	1 drop q3d	0	1	1	7

DBD: Diamond burr debridement; ID#: Identification number; KCS: Keratoconjunctivitis sicca or dry eye syndrome; OD (*oculus dexter*): Right eye; OS (oculus sinister): Left eye; OU (*oculus uterque*): Both eyes; PRA: Progressive retinal atrophy, q3d: *Quaque 3 die* = every 3 days; Y/N: Yes (1) or no (0).

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
