# Peer review of "Case Reports for Topical Treatment of Corneal Ulcers with a New Matrix Therapy Agent or RGTA^®^ in Dogs"

_vetsci, 2019, doi:10.3390/vetsci6040103_

Round 1

Reviewer 1 Report

There are still numerous areas of very awkward phrasing and grammar present in the corrected manuscript. The quality of the presentation would benefit very much from that "attention to detail" of having these areas addressed. All of the material is present, just that the way it is communicated makes the reader work harder to get it and quit before they comprehend what you have. I feel this information needs to be presented to the veterinary community in the best possible light. Good job on the rewrites.

Author Response

Reviewer 1

Comments and Suggestions for Authors

There are still numerous areas of very awkward phrasing and grammar present in the corrected manuscript. The quality of the presentation would benefit very much from that "attention to detail" of having these areas addressed. All of the material is present, just that the way it is communicated makes the reader work harder to get it and quit before they comprehend what you have. I feel this information needs to be presented to the veterinary community in the best possible light. Good job on the rewrites.

Answer: We thank the reviewer for the comments. A native English speaker and scientific writer has edited the phrasing and the grammar of this manuscript as required. Modifications are highlighted in green and bold for follow-up in the new submitted version of the manuscript.

Reviewer 2 Report

This clinical case report describes the treatment of 11 dogs at a single clinic with corneal defects/ injury with Clerapliq, a dextran derivative with HA-like properties.

The results suggest safety of the new treatment; it did not produce unexpected adverse events and was well-tolerated by the dogs and clients.

The results found that 83% of the cases responded to treatment over a period of 6-35 days. Because this is a case report and not a randomized, double blind, controlled clinical trial, this is not strong medical evidence of clinical effects. These results, however, are impressive since these dogs had chronic corneal issues which were not responsive to standard treatments. 

Critique:

This paper meets the description of a case report:  it is a short report detailing the signs, diagnosis and treatment prior to and after the administration of the new therapeutic agent. Concerns:  The agent used is labeled in EU.  It is clear whether this agent is registered with the FDA or whether an INAD is needed to test novel therapeutics in the USA. English language editing.  The paper has needs English language editing.  

Author Response

Reviewer 2

This clinical case report describes the treatment of 11 dogs at a single clinic with corneal defects/ injury with Clerapliq, a dextran derivative with HA-like properties.

The results suggest safety of the new treatment; it did not produce unexpected adverse events and was well-tolerated by the dogs and clients.

The results found that 83% of the cases responded to treatment over a period of 6-35 days. Because this is a case report and not a randomized, double blind, controlled clinical trial, this is not strong medical evidence of clinical effects. These results, however, are impressive since these dogs had chronic corneal issues which were not responsive to standard treatments. 

Critique:

This paper meets the description of a case report:  it is a short report detailing the signs, diagnosis and treatment prior to and after the administration of the new therapeutic agent.

Concerns:  The agent used is labeled in EU.  It is clear whether this agent is registered with the FDA or whether an INAD is needed to test novel therapeutics in the USA.

Answer: The agent used (Clerapliq®) in these case reports has already been used in dogs in USA but with human equivalent (Cacicol®) by the Department of Veterinary Clinical Sciences, College of Veterinary Medicine, Iowa State University, Ames, Iowa 50011, USA and the results published in the The Veterinary Journal, 2019 by Sebbag L et al, 233, p63-65b(https://doi.org/10.1016/j.tvjl.2018.01.003).

In our manuscript, the owners had signed the consents as mentioned in the manuscript to test Clerapliq® in their dogs in accordance with the relevant Louisiana state and USDA law and regulation.

This has been clarified lines 129-130, and by adding a specific paragraph lines 298-300:

Ethics approval and consent to participate: Written informed consent to treat the dogs with Clerapliq® was obtained from all owners when they attempted to the Eye Center for Animals Inc. (524 Moss Street, New Orleans, LA, USA) in accordance with the relevant Louisiana state and USDA law and regulation.

English language editing.  The paper has needs English language editing.  

Answer: We thank the reviewer for the comment and a native English speaker and writer has edited the phrasing and the grammar of this manuscript as required. Modifications are highlighted in green and bold for follow-up in the new submitted version of the manuscript.